# Provider and administrator-level perspectives on strategies to reduce fear and improve patient trust in the emergency department in times of heightened immigration enforcement

Carolina Ornelas-Dorian[1,2]*, Jacqueline M. Torres[3], Jennifer Sun[4], Alexis Aleman[5], Emmanuel Cordova[5], Aristides Orue[5], Breena R. Taira[5], Erik Anderson[4], Robert M. Rodriguez[1]

1 Department of Emergency Medicine, University of California San Francisco, San Francisco, California, United States of America, 2 School of Public Health, University of California Berkeley, Berkeley, California, United States of America, 3 Department of Epidemiology and Biostatistics, University of California San Francisco, San Francisco, California, United States of America, 4 Department of Emergency Medicine, Highland Hospital - Alameda Health System, Oakland, California, United States of America, 5 Department of Emergency Medicine, Olive View - UCLA Medical Center, Sylmar, California, United States of America

* carolina.ornelas@ucsf.edu

**Data Availability Statement:** All relevant data are within the paper and its Supporting information

## Abstract

### Study objectives

Heightened immigration enforcement may induce fear in undocumented patients when coming to the Emergency Department (ED) for care. Limited literature examining health system policies to reduce immigrant fear exists. In this multi-site qualitative study, we sought to assess provider and system-level policies on caring for undocumented patients in three California EDs.

### Methods

We recruited 41 ED providers and administrators from three California EDs (in San Francisco, Oakland, and Sylmar) with large immigrant populations. Participants were recruited using a trusted gatekeeper and snowball sampling. We conducted semi-structured interviews and analyzed the transcripts using constructivist grounded theory.

### Results

We interviewed 10 physicians, 11 nurses, 9 social workers, and 11 administrators, and identified 7 themes. Providers described existing policies and recent policy changes that facilitate access to care for undocumented patients. Providers reported that current training and communication around policies is limited, there are variations between who asks about and documents status, and there remains uncertainty around policy details, laws, and jurisdiction of staff. Providers also stated they are taking an active role in building safety and trust and see their role as supporting undocumented patients.

files. Our data is deposited at the UCSF Library's open access repository: https://doi.org/10.7272/Q66971TV.

**Funding:** This study was supported in part by a grant from the University of California, San Francisco Research Evaluation and Allocation Committee (REAC) Moffitt Fund and Pratt Fund (https://medschool.ucsf.edu/research/investigator-resources/reac-grant). The funders had no role in study design, data collection and analysis, decision to publish, or preparation of the manuscript.

**Competing interests:** The authors have declared that no competing interests exist.

## Conclusions

This study introduces ED-level health system perspectives and recommendations for caring for undocumented patients. There is a need for active, multi-disciplinary ED policy training, clear policy details including the extent of providers' roles, protocols on the screening and documentation of status, and continual reassessment of our health systems to reduce fear and build safety and trust with our undocumented communities.

## Introduction

A growing body of research has pointed to the role of immigration and immigration status (status) as social determinants of health (SDoH) [1, 2]. One key mechanism by which immigration and status may influence health outcomes is through impacts on health care access and utilization. Immigrants have historically had less healthcare access relative to their U.S.-born counterparts, with undocumented immigrants having the lowest rates of access to care [3]. Programs like the Affordable Care Act exclude undocumented immigrants from marketplace health care plans and Medicaid expansion [4].

Over the past several decades, restrictive policies and heightened immigration enforcement [1] have also been linked to reduced access to care for immigrants and their families [5–8]. Acute declines in health care utilization were observed following the 2016 U.S. Presidential election and amidst the COVID-19 public health crisis, which have led to reports of fear among immigrants in receiving testing and treatment [9–13]. Even in sanctuary cities that have limited cooperation with federal immigration authorities [14], structural barriers to accessing specialty care and social services persist [15, 16]. Racism and anti-immigrant discrimination contribute to these barriers and to fear and mistrust [1, 2, 13].

The ED serves as a safety-net provider for vulnerable populations, including patients who are undocumented or have poor access to primary care [3, 17]. This is in large part due to EMTALA (Emergency Medical Treatment and Active Labor Act), a federal law which mandates the provision of emergency care to all patients, regardless of insurance or immigration status, and generally allows providers to protect patient confidentiality and privacy (in addition to HIPAA, a federal law that protects patient health information) including from law enforcement, except in certain situations such as a court order [18]. Nevertheless, multiple studies have documented substantial fear and delays in care among immigrant patients presenting to the ED, particularly in the context of increased immigration enforcement [19, 20].

Despite well-documented barriers to accessing health care, including ED care, for immigrant patients, little is known about system-level efforts to reduce these barriers through policies and practices within EDs [2]. In this study, we seek to address this critical knowledge gap with a multi-site qualitative analysis of ED provider and administrator perspectives on policies and protocols to enhance trust and reduce barriers to ED care for immigrant patients in a time of heightened immigration enforcement. We focused on exploring and describing 1) the landscape of policies and protocols relevant to the care of immigrant patients at three safety-net EDs in California, 2) how ED providers and administrators in these settings experienced the creation, dissemination, and implementation of these policies and practices, 3) their evaluation of whether current practices are sufficient, and 4) how all of these domains varied across EDs.

## Methods

### Ethical statement

We obtained institutional review board (IRB) approval from the University of California San Francisco Committee on Human Research, the University of California Berkeley Committee on Protection of Human Subjects, the Olive View-UCLA Medical Center IRB, and the Highland Hospital—Alameda Health System IRB to conduct this study with written or verbal consent (recorded by researchers via a survey in our REDCap secure database). There were no minors included in the study.

### Study design and setting

From January 2020 to May 2020, we conducted a qualitative study at three urban safety-net hospitals in California: Zuckerberg San Francisco General Hospital (ZSFG) [San Francisco, CA], Olive View-UCLA Medical Center [Sylmar, CA] and Highland Hospital—Alameda Health System [Oakland, CA] with annual ED census of 73,000, 77,000, and 68,000 respectively. At each site, principal investigators led recruitment efforts using mainly a combination of in-person and scripted email recruitment, and occasional snowball sampling when individuals didn't respond to initial recruitment. We offered a $50 gift card for compensation.

Of 59 individuals recruited, 41 agreed to participate: 10 physicians, 11 nurses, 9 social workers, and 11 administrators. We conducted 17 interviews in-person and 24 interviews by phone during shelter-in-place orders for the COVID-19 pandemic. There were no significant differences in the interview themes conducted before and after the start of the COVID-19 pandemic. When COVID-19 was rarely mentioned, it was in the context of the importance of patients feeling safe to access medical care. Overall, it did not impact our analysis or interview content. Interview guides included questions covering respondents': awareness of existing policies—including both long-standing policies and those passed recently (within the last 5 years); understanding of the impact of immigration policies and enforcement on patient care; awareness of future policies planned within their ED; and ideas for other policies and practices that might be put into place to reduce fear and improve trust among immigrant patients in the ED (S1 and S2 Appendices). The questions were semi-structured and primarily open-ended. We pilot-tested the interview questions with a non-ED, senior physician and a focus group of four MPH students.

All interviewers were trained by the same lead researcher (CO) who, in addition to training, listened to select interview transcripts and provided direct feedback to interviewers. All but two participants consented to audio recording; for those two interviews, interviewers typed notes reflecting responses.

### Qualitative analysis

From March 2020 to May 2020, we conducted qualitative analysis. We analyzed data using Atlas.ti software and used constructivist grounded theory, which combines both deductive and inductive thematic analysis.

Two researchers (CO and JMT; one tied to emergency medicine and the other not) collaborated on the initial analytic steps by doing separate line-by-line analysis of the same interviews, comparing similarities and differences, and then summarizing the detailed codes into broader sub-groups and themes. In an iterative process, codes and themes were summarized into a codebook that provided definitions and inclusion/exclusion criteria. This codebook was revisited and researchers edited codes over time to incorporate novel codes and themes that emerged from the data. They applied the codebook to the same 5 additional interviews to

ensure consistency of coding practices. After discussing discrepancies, CO applied the codebook to the remaining interview data.

## Results

The 41 respondents included in our study were split evenly across all three EDs. Participants had substantial experience at their respective hospitals and within their roles. The median duration was 15 years for administrators and 10.5 years for clinicians.

Our coding yielded 360 individual codes, consolidated into 9 broad themes (S1 Table). We summarize key findings across each of the nine themes below.

Theme 1: *A majority of participants identified at least one of two longstanding hospital policies that help undocumented patients feel safe*. The two longstanding policies related to the care of undocumented patients in the ED are that "all staff do not have to cooperate and are under no obligation to release any information" (administrator) with immigration enforcement and that all patients are treated equally and respectfully in the ED, regardless of status (EMTALA and HIPAA). Providers also described other existing policies and protocols that they related to the care of immigrant patients in the ED, including not asking patients about their status and providing language services.

Theme 2: *Select participants reported a wide range of recent (past 5 years) policy and practice changes specific to undocumented patients in the ED that were responsive to heightened immigration enforcement and were driven by political factors, and patient and staff experience*.

At least one respondent at each site noted an increase in ED signage over the past five years with messaging "that are welcoming the undocumented individuals. . .to remove the fear factor" (social worker) in different languages. At one ED, multiple respondents were aware of a recent county-level policy that limits immigration enforcement and investigations at sensitive locations, like hospitals, without warrants or permits. Doctors and nurses at these sites felt these changes helped patients feel safer.

Others mentioned new initiatives. Some were specific to undocumented patients, such as not asking for identification until after patients see a medical provider, informing patients about changes in public charge rules (green card or visa denial for using government resources such as Medicaid; although it does not apply to all immigrants) [21], and increased informal conversations among staff about issues impacting undocumented patients. Other initiatives were more general, such as diversity and equity campaigns, increased social work capacity (e.g. 24-hour presence in the ED), and increased psycho-social resources for patients.

The most commonly cited factors that led to these policy and initiative changes were political rhetoric and actions (e.g. threat of immigration enforcement, deportation of a staff member), patient experiences (e.g. patients delaying care due to the fear of seeing immigration enforcement outside of the hospital), and staff initiatives (e.g. resident physician advocacy).

Theme 3: *There are limited previous and future policies specific to undocumented patients and administrators defend this*. Despite the two policy changes mentioned earlier, about a third of providers and administrators did not report any policy or protocol changes over the past five years specific to caring for undocumented patients, and indicated few to no proposed future changes. Administrators and a few providers across sites rationalized that their hospitals are "a community hospital of diversity and equity" and "we have always worked with the underserved" (administrator). Individual providers reported responding to each patient's unique needs, so there is less of an immediate need for specific policies or protocols.

Theme 4: *Current training on policies and patients' rights is limited. In general, communication about policies is top-down, email-based, and disparate*. At all sites, email from leadership was the main form of communicating existing policies, changes, or responses to federal threats

of immigration enforcement. However, some providers noted not remembering the exact email content and "wouldn't be able to give you a plan of action of what I'm going to do based upon a more in-depth discussion of immigration" (nurse). Over two-thirds of administrators and three-quarters of providers noted "very minimal" or "no formal" training specific to issues related to caring for undocumented patients. Current training that is relevant to care for undocumented patients, but not specific, were training modules about patient privacy and confidentiality, cultural sensitivity, and trafficking. Also, there were differences in training based on roles. At one site, these trainings were based out of the medical residency, limiting access for nurses and social workers, while at another site, it occurred in weekly meetings for social workers. The only participants to mention any training specific to responding to immigration enforcement was a social worker and a social work administrator at the same site who attended a training in a social workers' meeting.

Theme 5: *There is significant uncertainty across all provider-types on policies, laws, and the roles of staff.* Across all sites and roles, many were uncertain about existing policies and policy changes and/or their roles. This wasn't always expressed as "I don't know" and was not mutually exclusive with knowledge of policies. Uncertainties included the impact and communication of policies with patients, other specialties' communication, training, and role in caring for undocumented patients (e.g. registration/eligibility), the level of safety that providers can offer ("what am I allowed to do legally?" (nurse)), especially as it relates to the jurisdiction of law enforcement (e.g. the true meaning of a "sanctuary city"), and additional actions to take to protect patients. To address these uncertainties, a few providers noted relying on social workers to know the intricacies of these policies, while social workers admitted they had questions given their minimal training. Unprompted, a few participants confidently outlined steps on how to respond to immigration enforcement (e.g. moving patients to a secure area, informing the administrator of the day or risk management).

Theme 6: *Status is asked about and documented inconsistently across sites and providers. Although status does not impact clinical care, it comes up in a handful of clinical scenarios.* About half of interviewees did not ask about status and were not aware of patients' status. For those that were aware, it differed based on role. At one site, only social workers were aware, "information could be communicated in a note, patient's undocumented; it can be verbal, over the phone" (social worker). While at other sites, nurses and doctors were also aware. Nurses tended to ask more directly, while doctors asked indirectly (e.g. asking about health insurance). Interviewees learned about status through patients' insurance type or lack of health insurance, social security information collected at registration at some sites, self-disclosure, noticing ankle bracelets (for patients under immigration enforcement custody), and making assumptions based on immigration history.

Most of the providers that were aware of status did not document status in the electronic health record. However, a few social workers and doctors documented status when medically relevant in the medical chart (but did not clarify exact language was used in the documentation). Providers presented a dichotomy of trying not to document status to protect patients and confirming that status did not impact access to or quality of care, while acknowledging that status can play an important role in certain clinical situations (e.g. interpersonal violence, discharge planning, and referral to psychosocial resources).

Theme 7: *Providers have cared for undocumented patients who have experienced barriers to accessing emergency care and have delayed care due to fear of discovery.* The most common concerns patients have expressed to providers at all sites were encountering police in and around the hospital and the expense of visiting the ED. "A woman came in with tooth pain. . .was in severe pain. . ..trying to understand why she hadn't come in sooner. . .she was like, yeah. I came here two days ago, and I saw immigration, a vehicle outside of the hospital parked

outside of the emergency department and it scared me so much that I went home" (physician). Other concerns included deportation, public charge, and reporting crimes/accessing resources due to needing to interact with law enforcement or government workers. About half of providers across sites have experienced patients delaying care (for reasons including fear of discovery, financial burden, and limited time) and acknowledged the negative impacts from delaying care on morbidity and mortality from preventable conditions.

Theme 8: *Providers describe efforts to build safety and trust and considered their roles as protecting and supporting undocumented patients. Some even independently learn about immigration policies/ethics.* There were three key strategies identified for building safety and trust across all three sites. First, providers reassured patients they knew or suspected were undocumented that their visits were confidential with statements like, "You're safe with me. You're safe here. . .my job is to protect you, so you're safe" (physician). Second, there is inclusive messaging, with posters and signs that reiterate that all are welcome in the ED. Third, providers offered culturally and linguistically sensitive care, including accessing interpretation when relevant. There was a great sense of pride, especially among nurses and administrators, that they don't report patients or call the police. Only one physician mentioned interacting with immigration enforcement, due to the patient being in custody, for the sole purpose of providing medical care.

Several providers and administrators offered a description of their role as providing support and resources for undocumented patients, including providing information about which resources may trigger an investigation by immigration enforcement. All three hospitals cited strong cultures of commitment to caring for vulnerable populations. Other strategies that were unique to the respective hospitals included one handing out cards that describe "knowing your rights" and offering medical-legal partnership in the ED/hospital, another having a social medicine team in the ED to address non-medical psycho-social concerns and barriers, and the other having additional clinic support (e.g. asylum clinic). A few physicians, a social worker, and an administrator took time outside of their jobs to learn about addressing ethical issues, including immigration policies, in the ED.

Theme 9: *Interviewees suggest a wide range of future changes ranging from individual to structural changes.* The most common suggestion, that everyone, especially doctors and nurses, at all sites agreed on was the need for increased patient communication through signage, community outreach ("even if the hospital has a big sign that says, "immigrants are welcome", there's still a lot of distrust" (physician)), and direct patient education that reinforces messages of safety and commitment to providing care regardless of one's status. The second most common suggestion from administrators and a few providers was the need for further training on topics of patient rights, the legal environment, and current research on the impact of immigration status on health.

Physicians emphasized the importance of building health systems that integrate SDoH (e.g. creating a "fast track" in the ED for patients that need more social, rather than medical, support, and offering clinics with services specific to the needs of immigrant patients). Social workers and social work administrators emphasized the need for more psycho-social and medical resources that are available to undocumented individuals. Nurses, in particular, mentioned continuing to recruit diverse, Spanish-speaking staff (acknowledging that not all undocumented individuals speak Spanish).

## Discussion

A growing body of research and media have reported substantial fear and anxiety among undocumented patients and families seeking to access medical care, especially in times of

heightened immigration enforcement. Because the ED can often be the only source of care for undocumented communities, it is crucial that the ED remains a safe and welcoming space. Preventing delays in presentation to the ED is particularly critical during the current COVID-19 global pandemic [9]. There is growing but limited literature examining the policies and protocols in place within health systems to mediate this fear [22], and there are no studies specific to the ED.

Addressing this gap, we interviewed multidisciplinary ED teams in three California safety-net EDs to assess the current landscape of policies and protocols to reduce fear and enhance trust among immigrant patients, as well as providers' awareness of and experience with these policies. The included settings serve large immigrant patient populations within cities that have had long standing "sanctuary" status limiting cooperation with federal immigration authorities [14] and in a state with the largest number of immigrant residents in the U.S. [23].

We found that despite high awareness of the two major policies that inform care for undocumented patients in the ED (EMTALA and HIPAA), providers reported that current training and communication around policies is limited and disparate, and there remains uncertainty, across all roles and sites, around policy details, laws, and jurisdiction of staff. All sites have implemented relevant policy changes or new initiatives within the past five years in response to heightened immigration enforcements, but the majority of participants were not aware of these recent changes. There are few to no future proposed changes that are specific to undocumented immigrant patients.

Nevertheless, on an individual level, providers take an active role in building safety and trust, and some view their role as providing support and resources to undocumented patients. Providers have experienced patients delaying accessing emergency care for fear and non-fear-based reasons. Status impacts patient care, such as in cases of domestic violence and enrollment in social services, and there are variations among providers and sites between who asks about and documents status.

Reviewing the literature, we found only one other study examining system-level policies and practices for immigrant patients from the perspective of providers and administrators. Saadi and authors conducted in-depth interviews with clinicians, administrators, and senior executives. They compiled the range of hospital health system efforts across 5 states, including California, with established welcoming policies and strategies to mitigate perceived risk among undocumented immigrants, and have developed a toolkit, which includes strategies on reducing the risk around immigration enforcement and status disclosure, addressing patient and provider stressors (e.g. medical-legal partnerships, staff training), and coordination around risk mitigation [24]. However, most of these study sites were at Federally Qualified Health Centers or private or academic centers. Our study therefore complements this prior research with a focus on safety-net Eds [20] and is not limited to locations with established welcoming policies; we explore the general landscape of systems approaches, including barriers to implementation.

Our research leads us to several recommendations related to ED efforts to reduce fear and enhance trust among immigrant patients, and expand current efforts to build immigration-informed EDs, defined as recognizing immigration as a SDoH, and implementing immigration-informed interventions and advocacy [25] (see Table 1).

First, there is a need for further active, proactive, multi-disciplinary training on ED policies impacting care for undocumented patients, in addition to email communication (e.g. drills and scenarios) in order for staff to be prepared well ahead of critical times [20]. It's important that this training incorporates anti-racism so providers can critically engage with and address how racism and anti-immigrant discrimination impacts policies and patient experience [26]. Second, training and communication to staff about policies and protocols need to include

**Table 1. Policy and protocol recommendations for Emergency Departments (EDs) caring for immigrant patients.**

| Recommendation[a] | Description | Examples |
|---|---|---|
| Increase multi-disciplinary staff training on anti-racism and ED policies impacting care for undocumented patients. | Active policy and anti-racism training, in addition to email communication, that is proactive, not reactive to events that have already taken place. | Drills and scenarios to train staff on responses to protect patients in the case of immigration enforcement visiting hospitals [20]. |
| | | Anti-racism vocabulary and framework in collaboration with experts [26]. |
| Increase staff training and communication with policy and protocol details that clarify the jurisdiction and roles of providers, especially in relation to law enforcement. | Training and communicating the extent to which providers can offer confidential and safe care in and around the hospital, the roles of other healthcare staff (e.g. registration), and clear steps on how to respond to immigration enforcement visiting the hospital. | A multi-sector response team for immigration enforcement visits with a point person or team to stay up-to-date on policy changes and hospital best practices. |
| | | Easily accessible reference material in the workspace. |
| Foster a policy environment that removes the burden of action on individual ED providers and patients. | Support extraordinary interpersonal efforts by institutionalizing individual best practices to reduce bias, increase consistency in care, and reduce the cognitive load of providers. | Scripts inspired by current provider language to inform *all* patients of their rights in the ED along different points of contact (e.g. intake, rooming, discharge). |
| | | Welcoming signage in *all* ED rooms in multiple languages at different levels of health literacy to reassure patients without requiring patients to disclose status. |
| Implement a systematic process for whether, when, and how to screen for and document patient status. | Status is an important SDoH and can affect cases such as with domestic violence and discharge planning. Many EDs have resources for undocumented patients. Some document it when medically or legally relevant (e.g. asylum). However, asking can increase fear and documenting can expose patients' identity. There's a need for thoughtful policies and protocols to keep patients safe while providing holistic care. | Clear details on when and when not to document status. If necessary, standardized ways to tactfully phrase questions about status that reassure patients. If documenting, sharing preferred language and how the information is kept secure from immigration enforcement. |
| Continuous systems improvement and sharing of best practices to address the SDoH[b] and structural inequity that impact our patients. | Capture the experiences of those who utilize and experience the system and learn from and lift up each other's clinical and systemic best practices. | Patient and staff surveys to continuously gather feedback. |
| | | Sharing best practices formally and informally, especially around medical-legal partnerships, diversity and equity campaigns, and inclusive messaging campaigns. |

[a]: Source: based on findings from provider and administrator qualitative interviews in 3 California EDs from 03/2020–05/2020.

[b]: SDoH: social determinants of health.

specific information regarding the extent to which providers can offer confidential and safe care in and around the hospital, details on the roles of other healthcare staff (e.g. registration), and clear steps on how to respond to immigration enforcement visiting the hospital. Third, hospital systems must foster a policy environment that removes the burden of action on individual ED providers and patients, such as adapting existing provider language into standardized scripts to inform *all* patients of their rights, and reinforcing messages of safety in the design of the ED with welcoming signage in *all* ED rooms.

Fourth, EDs must have a systematic process for whether or not to screen for patient status. Asking patients about their status can induce fear, and in general, the literature does not recommend documenting status in order to protect the identity of patients [22, 27]. However, in practice, physicians or, more commonly, social workers often do ask. Immigration status is an important SDoH and can impact care such as with domestic violence and discharge planning, and inform connection to critical social and health-related resources (e.g. medical-legal programs). Our findings show variation in practice with some providers documenting it when legally and medically relevant (e.g. documenting the need for asylum). If screening is going to occur, details on when and how (including standardized, sensitive language) to screen for and

document status are needed. Finally, to achieve long-term system change, EDs must capture the experiences of those who utilize and experience the system (e.g. via patient and provider surveys), learn from and lift up each other's best practices both formally and informally, and collectively address the SDoH and structural inequity that impact our patients, such as through medical-legal partnerships.

## Limitations

This study has several limitations and strengths. First, this study occurred in three safety-net EDs in sanctuary cities within a sanctuary state. This limits the transferability of our findings to all ED settings within and outside of the U.S. Providers in states without sanctuary settings may experience even larger gaps in policies, training, and knowledge. Although this study is specific to the U.S., several other countries have federal laws similar to EMTALA, where doctors must provide basic emergency care to anyone, including undocumented immigrants [28]. Our study offers recommendations that could be adapted in other states, countries, and health systems within California. In addition, California is the U.S. state with the largest immigrant population (in a country with many immigrants), and safety-net EDs in the state therefore serve as important sites for this research. This offers an opportunity for future research to examine non-California ED health systems.

Second, we captured some, but not all, possible provider perspectives and experiences through our sampling strategy, but did reach thematic saturation among participants. A strength of this study was that trusted, well-respected, staff members led recruitment of interdisciplinary ED team members, which supported participation of busy, frontline providers. We were intentional in incorporating a multi-disciplinary approach to highlight the diverse voices who are critical members of the patient care team.

In addition, interviewers had dual insider-outsider status and were not direct peers or supervisors of any participant, which fostered an open environment for discussion. All interviewers received equal training, and within sites, the same interviewer conducted most, if not all, interviews, providing interview consistency.

Finally, we qualitatively examined provider and administrators' awareness and experiences of policies and protocols to enhance trust among immigrant patients, but did not evaluate pre-existing perceptions of undocumented patients, implementation of these policies and protocols, or their impact on patients' experiences and outcomes.

In summary, this study introduces policy approaches to caring for undocumented patients in three California safety-net EDs, and reiterates and expands on health system strategies for building, implementing, and communicating policies and systems specific to building safety and trust among our undocumented communities in the ED.

## Supporting information

**S1 Appendix. Interview guide for providers.**
(DOCX)

**S2 Appendix. Interview guide for administrators.**
(DOCX)

**S1 Table. Themes with additional exemplar quotes from a qualitative study interviewing providers and administrators in 3 California Eds.**
(DOCX)

## Author Contributions

**Conceptualization:** Carolina Ornelas-Dorian, Jacqueline M. Torres, Jennifer Sun, Alexis Aleman, Breena R. Taira, Erik Anderson, Robert M. Rodriguez.

**Data curation:** Carolina Ornelas-Dorian, Jacqueline M. Torres, Jennifer Sun, Alexis Aleman, Emmanuel Cordova, Aristides Orue, Breena R. Taira, Erik Anderson, Robert M. Rodriguez.

**Formal analysis:** Carolina Ornelas-Dorian, Jacqueline M. Torres.

**Funding acquisition:** Carolina Ornelas-Dorian, Jacqueline M. Torres, Robert M. Rodriguez.

**Supervision:** Carolina Ornelas-Dorian.

**Writing – original draft:** Carolina Ornelas-Dorian.

**Writing – review & editing:** Carolina Ornelas-Dorian, Jacqueline M. Torres, Jennifer Sun, Alexis Aleman, Emmanuel Cordova, Aristides Orue, Breena R. Taira, Erik Anderson, Robert M. Rodriguez.

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
