## [Decision Letter · Decision Letter 0]

24 Feb 2021

PONE-D-20-28725

Provider and administrator-level perspectives on strategies to reduce fear and improve patient trust in the emergency department in times of heightened immigration enforcements.

PLOS ONE

Dear Dr. Ornelas,

Thank you for submitting your manuscript to PLOS ONE. After careful consideration, we feel that it has merit but does not fully meet PLOS ONE’s publication criteria as it currently stands. Therefore, we invite you to submit a revised version of the manuscript that addresses the points raised during the review process.

We look forward to receiving your revised manuscript.

Kind regards,

Muhammed Elhadi, MBBCh

Academic Editor

PLOS ONE

Journal Requirements:

2. Thank you for including your ethics statement:  "We obtained institutional review board (IRB) approval from the University of California San Francisco Committee on Human Research, the University of California Berkeley Committee on Protection of Human Subjects, the Olive View-UCLA Medical Center IRB, and the Highland Hospital—Alameda Health System IRB to conduct this study with written or verbal consent. ".   

Please provide additional details regarding participant consent. In the ethics statement in the Methods and online submission information, please ensure that you have specified what type you obtained (for instance, written or verbal, and if verbal, how it was documented and witnessed). If your study included minors, state whether you obtained consent from parents or guardians. If the need for consent was waived by the ethics committee, please include this information.

Reviewers' comments:

Reviewer's Responses to Questions

**Comments to the Author**

1. Is the manuscript technically sound, and do the data support the conclusions?

Reviewer #1: Yes

Reviewer #2: Yes

Reviewer #3: Yes

Reviewer #4: Yes

Reviewer #5: Yes

2. Has the statistical analysis been performed appropriately and rigorously? 

Reviewer #1: N/A

Reviewer #2: N/A

Reviewer #3: N/A

Reviewer #4: Yes

Reviewer #5: N/A

3. Have the authors made all data underlying the findings in their manuscript fully available?

Reviewer #1: Yes

Reviewer #2: Yes

Reviewer #3: Yes

Reviewer #4: No

Reviewer #5: Yes

4. Is the manuscript presented in an intelligible fashion and written in standard English?

Reviewer #1: Yes

Reviewer #2: Yes

Reviewer #3: Yes

Reviewer #4: Yes

Reviewer #5: Yes

5. Review Comments to the Author

Reviewer #1: Thank you for the opportunity to review this manuscript. It is excellent. The authors provide a thorough overview of the challenges of ensuring the safety of undocumented immigrants in the ED, and their methods are well-described and appropriate. The only minor revisions that I believe this manuscript could use would be:

(1) a definition/explanation of the term "public charge," which may not be familiar to all readers

(2) including/incorporating 1-2 illustrative quotations per theme in the findings section. Although the themes are clearly described and this section is strong without the quotations, adding them to the text (rather than/in addition to having one per theme in Table 1) would add richness to the manuscript and presentation of findings.

(3) perhaps adding a reference to the toolkit developed from Saadi's study described on page 16, which includes many of the suggestions the authors describe (doctorsforimmigrants.com)

This paper presents an important contribution to the literature on the safety of undocumented immigrants in health care settings.

Reviewer #2: This is a worthwhile study and I appreciate your doing it. I do have a few questions and comments.

First, it appears that some of the authors are emergency physicians, who presumably have experience caring for undocumented people and likely came to this with some preconceptions and opinions. This should be acknowledged and discussed in the methods.

In the introduction, you say “These disparities were exacerbated under the Affordable Care Act due to the exclusion of undocumented immigrants from marketplace health care plans and Medicaid expansion.” This observation seems misleading as phrased because undocumented people were not harmed by the ACA, they just didn’t benefit from it. In fact some have family members, notably their children, who did benefit.

On page 6, you say you used a combination of in-person recruitment and snowball sampling. I don’t understand why you used snowball sampling. You evidently had the cooperation of the ED directors and presumably access to complete staff rosters. Why not use random or purposive sampling? It seems possible that respondents who referred others to you would have chosen like-minded people, which could affect the representativeness of your respondents.

On page 10 you write “informing patients about changes in public charge . . . “ I think I know what you mean but you should explain this.

Regarding the reference to ankle bracelets on page 12, I am guessing this refers to asylum seekers who have been released pending a hearing? You should explain. If so, ICE is already aware of these people and their status so there would not seem to be any particular danger to them from seeking care, although the public charge issue might be relevant, if I understand this correctly.

Regarding documentation of status, I note that I once worked for a community-based behavioral health provider that targeted a particular immigrant community. If we had to document status for some reason, we did it in a way that could not be detected even if the records were subject to a court order. If you can, you might make it more clear how status was documented in those cases where providers deemed it necessary.

On page 10 you write “Only one physician mentioned interacting with immigration enforcement, for the sole purpose of providing medical care.” Please explain. Why was it necessary to interact with immigration enforcement in order to provide medical care?

Regarding the limitations, why are nurses and social workers difficult to reach? In my experience it is must more difficult to get physicians to take the time to talk with me. I have no trouble interviewing nurses or social workers.

I like the tables, but make sure to format so they fit on one page.

Reviewer #3: Thank you for the opportunity to review this manuscript describing a qualitative research project conducted in three California emergency departments. The focus of the project is to explore the experience of undocumented patients and how staff can interact with them to provide care while mitigating fear and anxiety. The study is conducted among 41 participants including a range of emergency Department providers as well as administration. The researchers take valuable measures to ensure that the information is gathered without any hint of threat or coercion and adhere to sound principles of qualitative research.

The main findings of this work highlight 9 themes that cover a range of topics including existing policies that address undocumented patients as well as gaps in educational programs that optimize care for this vulnerable and marginalized population. The project goes on to draw from these conclusions and generate useful guidance for next steps that can address the issues that were identified.

Overall, I think this is a very well written and informative manuscript. While based on the California experience its findings have applicability in numerous jurisdictions. I also commend the authors for working in an area that highlights in equities and doing so in wave one of the covid pandemic is a remarkable accomplishment.

As a reviewer coming from a Canadian perspective I do have a number of suggestions that could make the manuscript more relevant to the non US reader. For example are the paper does not provide clear information related to how undocumented patience might be treated any differently in an emergency department than those who are legal. More information as to the range of complaints and pathologies that will bring an undocumented patient to the emergency department and which are unique from the general population would be appreciated. Similarly additional insight into the experience of the undocumented emergency department patient would provide useful additional context. Finally a more elaboration on US specific policies for example hippa and emtala might make the manuscript more accessible to the global reader.

Reviewer #4: This is an interesting work for health policy makers.

I have some concerns:

1- How did COVID-19 pandemic situation affected the study and the responses of ED individuals to your interview questions? I have a concern that these results might be biased by the COVID-19 situation.

2- How did you select the recruited individuals? Because without random sampling, this sample will not be representative.

3- How your findings will differ from other states and other countries?

Overall, the work is good and publishable.

Reviewer #5: This is a nicely written, qualitative piece which examines ED strategies to reduce fear and improve patient trust in times of heightened immigration enforcements. The study is unique and is important to form ED practice.

Significance: Is large given vulnerability of immigrant populations, particularly in the times of the pandemic,

Methods: While there were 40 subjects, there was a nice mix of different types of practitioners and the authors obtained saturation of themes.

Generalizablity: This is a qualitative survey of administrators, nurses and physicians in 3 safety net hospitals in California that care for a large number of immigrants. Although California has unique laws for immigrants, it also has the largest proportion of immigrants. This study can inform other California sites about strategies to help immigrants most readily but also can provide information to other EDs interested in this advocacy.

6. PLOS authors have the option to publish the peer review history of their article (what does this mean?). If published, this will include your full peer review and any attached files.

Reviewer #1: No

Reviewer #2: **Yes: **M. Barton Laws, Ph.D.

Reviewer #3: No

Reviewer #4: No

Reviewer #5: No

---

## [Author Response · Author response to Decision Letter 0]

2 Apr 2021

Thank you for your interest in our manuscript and for the informative comments. We greatly appreciate the careful review by the reviewers with excellent comments and suggestions. We have addressed the concerns in detail in our document "response to reviewers" and revised our manuscript accordingly. Thank you for your time and consideration.

---

## [Decision Letter · Decision Letter 1]

17 May 2021

PONE-D-20-28725R1

Provider and administrator-level perspectives on strategies to reduce fear and improve patient trust in the emergency department in times of heightened immigration enforcements.

PLOS ONE

Dear Dr. Ornelas,

Thank you for submitting your manuscript to PLOS ONE. After careful consideration, we feel that it has merit but does not fully meet PLOS ONE’s publication criteria as it currently stands. Therefore, we invite you to submit a revised version of the manuscript that addresses the points raised during the review process.

We look forward to receiving your revised manuscript.

Kind regards,

Muhammed Elhadi, MBBCh

Academic Editor

PLOS ONE

Additional Editor Comments (if provided):

The reviewers asked you to revise the manuscript and to address some other comments. In addition, you need to revise the paper by using English editing service or professional expert to ensure consistency and clarity of the manuscript before further consideration. 

Reviewers' comments:

Reviewer's Responses to Questions

**Comments to the Author**

1. If the authors have adequately addressed your comments raised in a previous round of review and you feel that this manuscript is now acceptable for publication, you may indicate that here to bypass the “Comments to the Author” section, enter your conflict of interest statement in the “Confidential to Editor” section, and submit your "Accept" recommendation.

Reviewer #1: All comments have been addressed

Reviewer #2: All comments have been addressed

Reviewer #5: All comments have been addressed

Reviewer #6: All comments have been addressed

2. Is the manuscript technically sound, and do the data support the conclusions?

Reviewer #1: Yes

Reviewer #2: Yes

Reviewer #5: Yes

Reviewer #6: Yes

3. Has the statistical analysis been performed appropriately and rigorously? 

Reviewer #1: N/A

Reviewer #2: N/A

Reviewer #5: Yes

Reviewer #6: Yes

4. Have the authors made all data underlying the findings in their manuscript fully available?

Reviewer #1: Yes

Reviewer #2: Yes

Reviewer #5: Yes

Reviewer #6: Yes

5. Is the manuscript presented in an intelligible fashion and written in standard English?

Reviewer #1: Yes

Reviewer #2: Yes

Reviewer #5: Yes

Reviewer #6: Yes

6. Review Comments to the Author

Reviewer #1: The authors have satisfactorily addressed all of my comments. Just two notes:

1. I disagree with many of the comments from Reviewer 2, and I believe some of the edits the authors made to address R2's concerns did not improve the paper.

2. In the limitations section, the authors comment on the generalizability of their findings, but it may be worth noting that qualitative work rarely aims at generalizability but rather at transferability.

Reviewer #2: (No Response)

Reviewer #5: Nice job on an important subject. Well written with unique methodology. I had minimal comments which have been addressed. The manuscript has been made even more relevant and generalizable internationally.

Reviewer #6: Carolina Ornelas and colleagues reported a study on "Policies and protocols on reducing undocumented patients’ fear in the emergency department”. They recruited and interviewed 41 ED providers and administrators from three California Eds, and found out existing policies and recent policy changes that facilitate access to care for undocumented patients; limited training and communication around policies; significant uncertainty across provider-types on policies, laws, and the roles of staff; providers stated they are taking an active role in building safety and trust and see their role as supporting undocumented patients. The authors conclude that there is a need for active, multi-disciplinary ED policy training, clear policy details including the extent of providers’ roles, protocols on the screening and documentation of status, and continual reassessment of our health systems to reduce fear and build safety and trust with our undocumented communities. Overall, the manuscript is professionally written and the study is designed well. However, I have one concern in policy and protocol recommendations that I will try to explain below.

Nearly a quarter of the nation’s undocumented immigrants reside in California, most of them are from Latin America, most of the others are from Asia. Multiple studies suggested that Latino immigrants facing the most discrimination in the USA, and COVID-19 fueling anti-Asian racism. Undocumented immigrants are even more vulnerable to experiences of discrimination under COVID-19 pandemic. Racism and Discrimination has been identified as a major stressor and influence on immigrant health. Undocumented immigrants are fearful of seeking medical attention because of not only the risk of exposing them to immigration enforcement but also being stigmatized and discriminated against by nonimmigrant-friendly medical providers. I think anti-racism and discrimination education of the training is needed in protocol recommendation, and could you please add this aspect in the discussion.

7. PLOS authors have the option to publish the peer review history of their article (what does this mean?). If published, this will include your full peer review and any attached files.

Reviewer #1: No

Reviewer #2: **Yes: **M. Barton Laws, Ph.D.

Reviewer #5: No

Reviewer #6: **Yes: **Hang Xing

---

## [Author Response · Author response to Decision Letter 1]

30 Jun 2021

In response to reviewer 1: Thank you for this comment. We have changed the language in the limitations section to reflect transferability, rather than generalizability; and have responded to comments from all reviewers, including reviewer 2

In response to reviewer 5: Thank you for this comment.

In response to reviewer 6: Thank you for this recommendation. We have included brief commentary on anti-racism training in the background, discussion, and policy recommendations.

---

## [Decision Letter · Decision Letter 2]

2 Aug 2021

Provider and administrator-level perspectives on strategies to reduce fear and improve patient trust in the emergency department in times of heightened immigration enforcement.

PONE-D-20-28725R2

Dear Dr. Ornelas-Dorian,

We’re pleased to inform you that your manuscript has been judged scientifically suitable for publication and will be formally accepted for publication once it meets all outstanding technical requirements.

Kind regards,

Muhammed Elhadi, MBBCh

Academic Editor

PLOS ONE

Additional Editor Comments (optional):

Reviewers' comments:

Reviewer's Responses to Questions

**Comments to the Author**

1. If the authors have adequately addressed your comments raised in a previous round of review and you feel that this manuscript is now acceptable for publication, you may indicate that here to bypass the “Comments to the Author” section, enter your conflict of interest statement in the “Confidential to Editor” section, and submit your "Accept" recommendation.

Reviewer #2: All comments have been addressed

Reviewer #5: All comments have been addressed

Reviewer #6: All comments have been addressed

2. Is the manuscript technically sound, and do the data support the conclusions?

Reviewer #2: Yes

Reviewer #5: Yes

Reviewer #6: Yes

3. Has the statistical analysis been performed appropriately and rigorously? 

Reviewer #2: N/A

Reviewer #5: Yes

Reviewer #6: N/A

4. Have the authors made all data underlying the findings in their manuscript fully available?

Reviewer #2: Yes

Reviewer #5: Yes

Reviewer #6: Yes

5. Is the manuscript presented in an intelligible fashion and written in standard English?

Reviewer #2: Yes

Reviewer #5: Yes

Reviewer #6: Yes

6. Review Comments to the Author

Reviewer #2: (No Response)

Reviewer #5: I have reviewed this manuscript now 3 times. I view the manuscript as being a contribution to the field

Reviewer #6: (No Response)

7. PLOS authors have the option to publish the peer review history of their article (what does this mean?). If published, this will include your full peer review and any attached files.

Reviewer #2: **Yes: **M. Barton Laws, Ph.D.

Reviewer #5: No

Reviewer #6: No

---

## [Editor Report · Acceptance letter]

3 Sep 2021

PONE-D-20-28725R2 

Provider and administrator-level perspectives on strategies to reduce fear and improve patient trust in the emergency department in times of heightened immigration enforcement. 

Dear Dr. Ornelas-Dorian:

I'm pleased to inform you that your manuscript has been deemed suitable for publication in PLOS ONE. Congratulations! Your manuscript is now with our production department. 

Kind regards, 

on behalf of

Dr. Muhammed Elhadi 

Academic Editor

PLOS ONE